# Management and Clinical Outcomes of Scleredema Diabeticorum: A Scoping Review

**DOI:** 10.3390/diseases13100346

**Published:** 2025-10-17

**Authors:** Weeratian Tawanwongsri, Chime Eden

**Affiliations:** 1Division of Dermatology, Department of Internal Medicine, School of Medicine, Walailak University, Nakhon Si Thammarat 80160, Thailand; 2Division of Dermatology, Jigme Dorji Wangchuck National Referral Hospital (JDWNRH), Thimphu 11001, Bhutan; chime.eden17@gmail.com

**Keywords:** scleredema adultorum, diabetes mellitus, disease management, treatment outcome, glycemic control

## Abstract

Background/Objectives: The objective of this scoping review was to systematically map the available evidence for the management of scleredema diabeticorum (SD) to summarize the documented clinical outcomes with the aim to inform clinical practice and identify research gaps. Methods: We conducted a scoping review identifying studies published in English from January 2005 to July 2025 through a comprehensive search of Scopus, MEDLINE, and the Cochrane Library. Eligible studies included randomized controlled trials, observational studies, case series, and case reports on treatment interventions and clinical outcomes. Two reviewers independently screened records, extracted data, and narratively and descriptively synthesized data. Results: Forty-five studies met the inclusion criteria: 39 single-patient case reports, five case series, and one multicenter observational study. The most common interventions were PUVA (8 studies, 14 patients; 12/14 improved, 85.7%), methotrexate (8 studies, 22 patients; 8/22 improved, 36.4%), and improved glycemic control (9 studies, 12 patients; mixed responses). Across small case reports/series, PUVA, UVA-1, and IVIG were most frequently reported as beneficial. Methotrexate monotherapy showed low and inconsistent effectiveness, with higher responses when combined with other agents. Other reported therapies included colchicine, electron-beam radiation, tranilast, and topical hyaluronidase. Conclusions: PUVA, UVA-1, and IVIG may offer benefit, while methotrexate alone is often ineffective. Evidence is predominantly from case reports and small series, which makes it difficult to generalize. Multicenter trials with standardized protocols are needed to develop evidence-based treatment recommendations.

## 1. Introduction

Scleredema of Buschke is a rare condition, and its exact incidence remains unknown [1]. It can be classified into three types. Type 1, which accounts for approximately 55% of cases, typically affects children and young adults and presents abruptly following a febrile illness, most commonly of streptococcal origin [2]. Type 2, representing approximately 25% of cases, has a gradual onset and is associated with hematologic disorders such as paraproteinemia, monoclonal gammopathy, multiple myeloma, and amyloidosis [3]. Type 3, or scleredema diabeticorum (SD), accounts for approximately 20% of cases and is observed in individuals with poorly controlled, long-standing insulin-dependent diabetes. Its pathogenesis may involve the non-enzymatic glycosylation of dermal collagen [4]. SD is a subtype of scleredema characterized by diffuse, symmetrical induration, and thickening of the skin owing to mucin deposition in the dermis [5]. It is more commonly observed in middle-aged and obese adults with a higher prevalence in men. It typically affects the upper back, posterior neck, shoulders, and sometimes the chest [6]. Although skin changes are generally painless, they can cause significant stiffness and reduced mobility. Some patients report pain and tightness in the affected areas, which can severely impair their range of motion and interfere with daily activities. In severe cases, skin thickening may extend to the trunk, potentially leading to restrictive lung disease and associated respiratory symptoms [7].

The diagnosis is primarily clinical and supported by histopathological findings. Laboratory and immunological investigations are generally not required. Skin biopsy is useful for excluding scleroderma-like conditions. Histopathology typically reveals marked thickening of collagen bundles in the middle and reticular dermis, separated by clear mucin-filled spaces [8,9]. Mild mononuclear inflammatory infiltrates may be observed around the capillaries. Subcutaneous fat often shows invasion by enlarged collagen fibers, whereas the epidermis, sweat glands, and pilosebaceous units remain unaffected. Mucin deposition is consistent in most of the cases. Differential diagnoses include several scleroderma-like conditions [10]. Systemic scleroderma is a connective tissue disease that can cause skin thickening but is usually accompanied by additional systemic symptoms. Eosinophilic fasciitis also presents with skin induration and is often associated with peripheral eosinophilia but differs in histological appearance. Scleromyxedema, commonly linked to monoclonal gammopathy, is characterized by papular skin lesions and distinct histopathological features. Amyloidosis may also cause cutaneous changes, although it typically manifests with other systemic signs that differentiate it from scleredema.

The management of SD involves a multifaceted approach [11]. Optimizing diabetes control is essential; tight glycemic control and weight management have been shown to improve symptoms [3]. Physical therapy, which includes regular physiotherapy and exercise, helps maintain joint mobility and reduces skin stiffness. Phototherapy, particularly ultraviolet A1 (UVA-1), has demonstrated significant clinical benefits and is often preferred, whereas psoralen plus ultraviolet A (PUVA) therapy may be effective in some cases [12]. Topical corticosteroids may be used for localized symptoms, while systemic corticosteroids or immunosuppressants are reserved for severe or refractory cases [9].

To date, the majority of available evidence for SD management has been limited to individual case reports and small series, which involve a range of treatment regimens and outcome measures. There has been no prior scoping review that has systematically mapped the full range of interventions and clinical outcomes for SD. Previous publications have primarily focused on an individual treatment methodology or a case experience with only a limited understanding of the overall therapeutic landscape. This study has sought to fill the gap in the literature by synthesizing and summarizing all available evidence on the management and clinical outcomes for SD. The aim of this scoping review was to systematically map the existing literature, evaluate reported therapeutic efficacy and safety, and identify knowledge gaps to inform future research and clinical decision-making.

## 2. Materials and Methods

This scoping review was conducted following the PRISMA 2020 guidelines and the Joanna Briggs Institute’s (JBI) methodology for scoping reviews [13]. The protocol was retrospectively registered with the International Platform of Registered Systematic Review and Meta-Analysis Protocols (INPLASY; registration number INPLASY202570116; https://inplasy.com/inplasy-2025-7-0116/, accessed on 29 July 2025). The manuscript outlines any deviations from the protocol with justification in the appropriate section. This study was conducted in accordance with the Declaration of Helsinki and was submitted for ethics review prior to data extraction and subsequently approved by the Walailak University Ethics Committee (protocol code WUEC-25-309-01, date of approval: 15 August 2025).

### 2.1. Search Strategy

The search strategy aimed to identify published studies reporting the management of SD and associated clinical outcomes. A three-step search was performed in accordance with the JBI methodology. First, a preliminary search of MEDLINE (via PubMed) and Scopus was conducted to identify relevant articles. The text words contained in the titles and abstracts, along with the index terms, were analyzed to inform the development of a comprehensive search strategy. The final search was performed in Scopus, MEDLINE, and the Cochrane Library using a combination of keywords and Boolean operators.

(“scleredema diabeticorum” OR (“scleredema” AND “diabetes”)).

The search was limited to studies published in English between January 2005 and July 2025 to capture contemporary evidence while excluding outdated treatment approaches. Only peer-reviewed articles were included. Eligible study designs comprised randomized controlled trials, prospective and retrospective studies, case reports, case series, and letters. Studies that did not report treatment interventions or clinical outcomes were excluded along with reviews, conference papers, book chapters, books, and non-peer-reviewed materials. All identified records were exported to a reference management tool, and duplicates were removed. Two reviewers independently screened the titles and abstracts, followed by a full-text review based on the predefined eligibility criteria. Reference lists of the included studies were examined using additional relevant sources. The PubMed search strategy is presented here as an example and has been adapted for use in other databases. The PubMed search strategy is exemplified and subsequently adapted for use in Scopus and the Cochrane Library by modifying the field tags and controlled-vocabulary terms to reflect the indexing conventions for each database. The adapted search strings for all databases are offered in Appendix A and are provided for transparency and reproducibility.

(“scleredema diabeticorum”[Title/Abstract] OR (“scleredema”[Title/Abstract] AND “diabetes”[Title/Abstract])).

### 2.2. Source of Evidence Selection

All records identified through the database searches were imported into Microsoft Excel, and duplicates were removed. Pilot screening was conducted to ensure consistency in the eligibility criteria. Two reviewers independently screened the titles and abstracts of the remaining articles. The full texts of potentially relevant articles were retrieved and assessed in detail by the same two reviewers. Any disagreements during the screening or selection processes were resolved through discussion. The reasons for excluding studies at the full-text stage were recorded and reported in the final review. The selection process was conducted in accordance with the PRISMA-ScR (Preferred Reporting Items for Systematic Reviews and Meta-Analyses extension for Scoping Reviews) guidelines and illustrated in a PRISMA flow diagram, as shown in Figure 1 [14].

### 2.3. Data Extraction

Two reviewers independently extracted data from the included studies, using a standardized data extraction form developed specifically for this particular review. The information to be extracted from each study, which was recorded within the form, included author and year of publication, country of study, patient characteristics (e.g., age, sex, comorbidities, type of diabetes, baseline glycemic status, e.g., HbA1c or fasting glucose, body weight or BMI, insulin regimen), disease onset/duration, details of the treatment, and reported outcomes/clinical response.

A draft extraction form was piloted on a subset of included studies to ensure clarity and relevance. Revisions were made as required during the data extraction process, and any modifications were documented in the final review. Any discrepancies between reviewers were resolved through discussion. Where necessary, authors were contacted to clarify or obtain missing information. Consistent with the JBI methodology, a critical appraisal of individual sources of evidence was not performed, because this step is not required for scoping reviews.

### 2.4. Data Charting and Synthesis of Results

The extracted data are presented in both tabular and narrative forms to address the review objective and research questions. Key findings such as patient characteristics, treatment modalities, and clinical outcomes were organized into summary tables, allowing for a clear comparison across studies. Where appropriate, additional visual formats (e.g., bar charts or frequency tables) were employed to illustrate the distribution of interventions and outcomes.

A narrative synthesis accompanied the tabulated results, providing contextual interpretation and highlighting patterns and trends in treatment approaches and clinical responses. This summary also outlined the contributions of the findings to the current understanding of SD management and identified knowledge gaps to guide future research.

## 3. Results

### 3.1. Characteristics of Included Studies

A total of 45 studies were included, comprising 39 single-patient case reports, five case series, and one multicenter observational study (Table 1). Publications spanned 2005–2025 and represented at least 20 countries across Europe, Asia, North America, and Africa, with Spain (five studies), the United States (four studies), and Japan (three studies) being the most frequently represented. Most patients were middle-aged (40–70 years), with diabetes duration ranging from new-onset to more than 35 years. One pediatric case described a 13-year-old girl with recurrent SD. Of the studies that reported diabetes type, 33/41 (80.5%) involved type 2 diabetes mellitus and 8/41 (19.5%) involved type 1. Monoclonal gammopathy was reported in three cases. Poor glycemic control with microvascular complications was common, and many patients had obesity or metabolic comorbidities; less frequent associations included monoclonal gammopathy, autoimmune disease, and restrictive lung disease. Diabetes-related complications included microvascular complications, such as retinopathy, nephropathy, and neuropathy. Lesions most often affected the posterior neck, shoulders, and upper back, with occasional generalized or atypical distribution. Disease onset ranged from acute (10 days) to very gradual over more than 20 years, although most cases developed insidiously from months to several years, and the course was typically chronic (2–10 years), sometimes with recurrence.

### 3.2. Reported Therapies

The most frequently described interventions were phototherapy (PUVA and UVA-1), MTX, and optimization of glycemic control. Based on the included studies, PUVA was reported in 8 studies (14 patients) [3,6,8,25,35,40,51,52]––including one case combined with MTX [16]; UVA-1 in 7 studies (14 patients) [3,15,17,30,44,47,49]; MTX in 8 studies (22 patients) [3,16,17,21,22,35,39,53]; and improved glycemic control in 9 studies (12 patients) [8,18,20,32,37,38,41,43,50]. Less-common interventions included IVIG in 3 studies (4 patients) [21,28,31], electron-beam/radiation in 3 studies (3 patients) [17,22,46], colchicine in 4 studies (7 patients) [3,16,40,54], allopurinol in 2 studies (2 patients) [37,41], topical hyaluronidase in 1 study (1 patient) [26], tranilast in 1 study (3 patients) [29], and tamoxifen in 1 study (2 patients) [42].

### 3.3. Clinical Effectiveness

Across eight studies (14 patients), PUVA was associated with clinical improvement in 12 of 14 treated patients (85.7%)—most commonly softer cutaneous induration, reduced erythema, and improved range of motion—using oral methoxypsoralen with UVA two to three times per week for approximately 20–40 sessions, with cumulative dose reported in some series [3,6,8,25,35,40,51,52]. Seven studies (14 patients) indicated clinical improvement in 13 of 14 patients (92.9%) treated with UVA-1, including cases resistant to other modalities [3,15,17,30,44,47,49]. Typical regimens were 30–60 J/cm^2^ per session, three to five times weekly, for 12–24 sessions (cumulative dose 1200–1800 J/cm^2^).

Three case reports (three patients) described IVIG administered at 2 g/kg per cycle over two to five consecutive days at approximately four-week intervals for three to ≥10 cycles, with clinical benefit in all cases [21,28,31]. Clinical improvement was maintained while treatment continued, and in one case relapse occurred two to three months after discontinuation but improved again upon re-initiation [31].

Methotrexate produced limited overall effectiveness, 8 of 22 patients (36.4%) achieved at least partial improvement, whereas 14 of 22 (63.6%) had minimal or no benefit despite dose ranges of 5–25 mg per week (oral or subcutaneous) for ≥2 months to >1 year [16,17,21,22,35,39,53]. Partial responses were most clearly documented in a small prospective series [39] and in selected combination contexts [3,35], whereas several modern reports noted little to no benefit [16,17,21,22]. Dosing was typically initiated at 5–15 mg per week, with escalation as tolerated.

Clinical improvements were reported in isolated series for topical hyaluronidase (sustained improvement at 24 months in one patient) [26] and tranilast (improved mobility with reduced dermal thickness on imaging in three patients treated for approximately three months) [29]. Tamoxifen improved symptoms in two patients but was complicated by vaginal bleeding, leading to discontinuation and relapse after cessation [42]. Reports of colchicine and allopurinol were mixed or limited to single-patient experiences [3,40,41,43].

Case-level evidence suggests that electron-beam regimens of 20–24 Gy delivered in 10–12 fractions, or photon volumetric-modulated arc therapy at 20 Gy in 10 fractions, can provide symptomatic relief of induration and improve mobility [17,22,46]. Durability appeared shorter with lower-dose electron courses (e.g., 12 Gy in six fractions) compared with 20–24 Gy schedules; re-irradiation was reported as feasible in selected patients with careful field design and dose constraints [17].

### 3.4. Treatment Failure and Disease Progression

Despite the range of available therapies, nonresponse and disease progression were observed in SD. With methotrexate, 14 of 22 patients (63.6%) showed minimal or no clinical improvement across the included reports [16,17,22,39,53]. Colchicine had mixed efficacy; 3 of 7 patients (42.9%) across four studies showed no benefit, particularly when used as monotherapy [3,16,40,54]. Cyclosporine produced minimal or no response in all 3 of 3 patients (100.0%) reported across two studies [22,52]. Systemic corticosteroids failed to achieve clinical improvement in 4 of 7 patients (57.1%) [17,24,30,33]. Refractoriness was also described with multimodal regimens (combinations of immunosuppressive agents and phototherapy) in two studies [16,17]. Relapse after treatment withdrawal was documented in two patients—one following discontinuation of tamoxifen and one after cessation of intravenous immunoglobulin—with subsequent improvement upon re-initiation where reported [31,42]. A single fatal outcome related to acute respiratory failure was reported as a rare complication [48].

### 3.5. Adverse Events and Safety Considerations

Among the 45 studies reviewed, ten (22.2%) specifically acknowledged or discussed adverse events, 26 (57.8%) stated the treatments were well tolerated without reported adverse effects, and 9 (20.0%) discussed safety in a single sentence or did not discuss it. A brief table summarizing adverse events and safety considerations is available in Appendix A. Of the 10 studies with reported adverse events, six described mild, self-limited adverse events, such as transient erythema or tanning that occurred after phototherapy [3,6,8,44], a headache occurring either during or immediately after intravenous immunoglobulin therapy [31], and vaginal bleeding that was temporally associated with tamoxifen treatment [42]. One study reported a worsening of scleredema during insulin glargine administration, which improved after switching to regular insulin [20]. Two studies described severe events that were attributed to disease progression, rather than the treatment, specifically anhidrosis with heat intolerance [37] and a fatal cardiopulmonary complication [48]. Kiyohara and Tanimura conducted prospective safety monitoring of repeated hyaluronidase injections and reported no adverse events following repeated injections [26]. For clarity, studies that reported adverse events are categorized according to the therapeutic agent used, and the frequency and type of adverse effects are described in the following subsections.

#### 3.5.1. Intravenous Immunoglobulin (IVIG)

Three case reports evaluated IVIG at 2 g/kg per cycle given over 2–5 days at 4-week intervals for 3–≥10 cycles [21,28,31]. Clinical benefit was observed in all three patients. Adverse events occurred in 1/4 patients (mild headache); no serious systemic complications were reported. Responses were maintained while on treatment; one case relapsed 2–3 months after discontinuation and improved on re-initiation [31].

#### 3.5.2. Phototherapy

Across phototherapy reports, 7 of 15 studies (46.7%) reported minor, transient reactions in 12/79 patients (15.2%), generally limited to erythema, tanning, or a burning sensation [6,8,15,40,44,51,52]. The remaining reports described good tolerability without notable adverse effects [3,47,49]. No long-term phototoxicity, carcinogenesis, or photoaging was documented.

#### 3.5.3. Systemic and Oral Pharmacologic Therapies

Across the methotrexate reports, adverse events were not prominently described, where noted, treatment was well tolerated [3,35,39,53]. Systemic corticosteroids were used in six studies (20 patients). Two studies noted hyperglycemia or metabolic intolerance [3,22], whereas four studies reported no complications [30,32,33,51]. Tranilast was well-tolerated, with no adverse events reported. [29]. Allopurinol [41], colchicine [40], and aminobenzoate with dimethyl sulfoxide [54] were also reported without adverse effects. Tamoxifen was used in two cases within one study; both patients developed vaginal discharge, and one experienced vaginal bleeding that led to treatment discontinuation [42].

#### 3.5.4. Radiation and Device-Based Therapies

Electron-beam/radiation therapy (reported in two patients within this safety subset) was well tolerated, with no significant treatment-related toxicity recorded [46]. Frequency-modulated electromagnetic neural stimulation (FREMS) was used in one patient, and no adverse reactions were reported [36].

### 3.6. Glycemic Control and Diabetes Type

Across the included case reports, glycemic status and diabetes type were heterogeneous (Appendix A). Of the 41 studies that reported diabetes type, 33 (80.5%) involved type 2 diabetes mellitus and 8 (19.5%) involved type 1 diabetes mellitus; several reports noted overlapping or uncertain classification. Most patients had long-standing diabetes with chronically suboptimal control, with mean glycated hemoglobin (HbA1c) values typically 8.0–10.5%. The cutaneous response to improved glycemic management was variable: two studies documented partial improvement in induration following better HbA1c or changes in insulin therapy [32,42], whereas others reported little to no cutaneous improvement despite reasonable metabolic control [51,53]. Several case reports suggested that adjunctive therapies (e.g., phototherapy, tamoxifen) were associated with improvement while glycemic control was optimized, but the benefit appeared to extend beyond glycemic management alone. Collectively, these observations support that although optimization of diabetes care remains essential, cutaneous outcomes may be limited once dermal fibrosis is established.

## 4. Discussion

In this scoping review, the management of SD was characterized by considerable heterogeneity in therapeutic approaches, dosages, and treatment durations. Phototherapy, particularly PUVA and UVA-1, and IVIG were among the most consistently effective interventions, whereas MTX demonstrated variable and limited benefits. IVIG regimens ranged from 1.35 g/kg over 3 days to 2 g/kg every 4 weeks, with an outlier of 3 g/week for 3 cycles; treatment durations extended from a few cycles to more than 11 cycles, frequently in combination with phototherapy or other immunosuppressants, and all treated patients showed clinical improvement. Adverse events associated with IVIG therapy were mostly mild, including transient headache or palmar dyshidrosis, and there were no concerning systemic adverse events. Phototherapy was tolerated well overall, with only mild erythema, tanning, or burning post-treatment, with no evidence of long-term phototoxicity. MTX dosing ranged from 5 to 25 mg/week, administered orally or subcutaneously, with treatment periods ranging from 2 months to >1 year. Lower starting doses (5–15 mg/week) were often titrated upward according to the clinical response and sometimes combined with PUVA, phototherapy, or systemic corticosteroids. Despite these combinations, only eight of 22 MTX-treated patients achieved partial improvement, and most reports described minimal or no benefit, underscoring the need for more effective systemic options. The precise role of glycemic control in the SD patient population is indeterminate. Most patients with SD show signs of chronic hyperglycemia, which is characterized by insulin resistance; however, improvement in metabolic status was unable to counterbalance dermal thickening. Improvement has been documented in isolated case reports after improvement in glycemic control; however, it is clear that most cases will progress, independent of achieving wanted HbA1c levels. Incidentally, this variability demonstrates that any cutaneous manifestations will immediately persist after years of advanced glycation and fibrosis, rendering retrogression unlikely. Glycemic optimization remains an important component of holistic diabetes management and is an important strategy for delaying and potentially preventing disease progression.

The pathogenesis of SD is thought to be driven primarily by the chronic metabolic and vascular consequences of long-standing diabetes mellitus. Persistent hyperglycemia induces endothelial dysfunction through oxidative stress, inflammation, and altered vasoactive signaling, leading to diabetic microangiopathy, impaired capillary blood flow, and structural vessel damage [56,57]. This microvascular injury reduces tissue perfusion, creating a chronic hypoxic environment that perpetuates oxidative stress via excess reactive oxygen species, further exacerbating vascular injury and dermal hypoxia [58,59]. Concurrently, sustained hyperglycemia promotes the non-enzymatic glycation of collagen, resulting in the formation and accumulation of advanced glycation end products that irreversibly crosslink collagen fibers, increasing their stiffness, insolubility, and resistance to degradation [60,61]. These glycosylated and crosslinked collagen fibers progressively accumulate in the dermis, accompanied by increased mucin (glycosaminoglycan) deposition, which is likely promoted by local inflammatory mediators [62]. The combined excess of stiff collagen and mucin leads to characteristic dermal thickening and non-pitting induration of the upper back, neck, and shoulders, which clinically manifests as SD [3].

Evidence regarding whether improved glycemic control leads to the resolution or improvement of SD remains inconclusive, with reports showing mixed outcomes. Some cases have demonstrated notable clinical improvement following the optimization of blood glucose levels, whereas others have reported minimal or no change despite substantial reductions in HbA1c levels. Chatterjee [32] reported the case of a 61-year-old male in whom improved glycemic control and weight loss were associated with reductions in neck pain, stiffness, and skin induration over a 4-year period. Similarly, Verma et al. [20] documented the resolution of SD in a 13-year-old girl following 2 months of improved glycemic control, with subsequent recurrence responding again to insulin therapy. Mehta et al. [43] observed mild softening of lesions after 2 months of insulin therapy, and Baillot-Rudoni et al. [50] reported dramatic improvement in four patients with long-standing type 1 diabetes following intraperitoneal insulin therapy and HbA1c reduction. In contrast, other reports have described persistent lesions despite substantial improvements in glycemic control. Ranabahu et al. [19] found no improvement in a 49-year-old man after 6 months, despite an HbA1c reduction of 6.8%. Similarly, Shahzad [38] noted unchanged lesions after 1 year, despite lowering HbA1c from 10 to 11.5% to 8%, and Chatterjee et al. [27] reported no improvement in a female whose diabetes management remained suboptimal. These inconclusive responses to glycemic control may reflect the complex pathogenesis of the disease as well as heterogeneity in chronicity, severity, and individual patient factors. Further studies are needed to ascertain the prognostic factors that may predict which patients with SD will respond to better glycemic control, because our understanding of SD pathophysiology and its prognostic factors is still somewhat limited. It seems likely that several interrelated factors drive both the persistence of the disease as well as variation in the response to treatment. One treatment may not suffice to enable a clinically relevant response in patients with a complex or poor prognosis. These variables may help to explain the heterogeneous response seen among treatment modalities. Future studies on SD could delineate these factors, particularly the effects of long diabetes duration, established microvascular complications, obesity, and metabolic comorbidities on the response to treatment or the progression of the disease.

Phototherapy modalities reported for the treatment of SD include UVA (320–400 nm), UVB (280–320 nm), and combination regimens that utilize both wavelengths. UVA, owing to its greater dermal penetration, exerts direct effects on dermal fibroblasts by downregulating collagen synthesis and promoting collagenase activity, thereby counteracting the excessive collagen accumulation characteristic of the disease [63,64]. These mechanisms are particularly relevant in SD, where glycosylated and crosslinked collagen contribute substantially to dermal thickening. The reported efficacy of phototherapy for SD varies, with outcomes ranging from complete or partial improvement to no response. PUVA has demonstrated notable clinical benefits in several cases. Martín et al. [8] reported improved mobility, skin softening, and resolution of erythema in a 53-year-old male after 2 months of PUVA. Sarı et al. [6] observed partial resolution of erythema and edema, along with softer skin and improved mobility, after 2 months of local PUVA in a 54-year-old female. Kokpol et al. [40] documented marked softening (50% after 20 sessions and 80% after 40 sessions) and reduced dermal thickness following treatment with PUVA plus colchicine. In a series by Simó-Guerrero et al. [25], all five patients who received PUVA achieved partial improvement, although some patients experienced recurrence. Nakajima et al. [52] described two long-standing SD cases with marked clinical and histological improvements following oral PUVA. Conversely, Meguerditchian et al. [51] found no skin improvement with PUVA despite concurrent physiotherapy, and Mamadpur and Singh [16] reported no benefit in one patient treated with MTX plus PUVA.

UVA-1 phototherapy has also demonstrated promising outcomes. Kroft and De Jong [47] reported excellent or good clinical responses in all three treated patients, with improvements in skin softening and mobility confirmed through durometry. Similarly, Thumpimukvatana et al. [44] described >75% softening in one patient, sustained for 2 years, and relapse-responsive improvement in another. Lewerenz and Ruzicka [49] observed marked improvement within weeks in two of three patients. Llamas-Segura et al. [15] reported complete resolution in one patient and adequate improvement in two others without recurrence over 3–24 months. Linares-González et al. [23] documented improved dysphagia, mobility, and neck stiffness after 28 sessions of UVA-1 in a patient who was unresponsive to multiple systemic therapies.

Other phototherapeutic modalities have been investigated. Waqar et al. [5] reported symptomatic and functional improvement after 3 months of narrow-band UVB (NB-UVB). In contrast, Hong et al. [21] observed significant and sustained improvements in skin and mobility only when NB-UVB was combined with monthly IVIG, after NB-UVB monotherapy had proven ineffective. Electron-beam radiation has yielded partial to complete, but often temporary, improvement. Gracie and Whitaker [17] reported transient complete responses followed by relapse, and Kyriakou et al. [22] observed partial improvement after failure of multiple systemic therapies. Yu et al. [46] noted sustained symptom relief but no imaging changes after 20 Gy of electron beam therapy. Overall, PUVA and UVA-1 appear to be most consistently beneficial, particularly in combination regimens, although responses are inconsistent and relapses remain common.

IVIG has demonstrated efficacy in the treatment of SD, primarily through its immunomodulatory and antifibrotic effects [31]. By modulating immune responses, IVIG may attenuate the chronic inflammation implicated in the disease’s pathogenesis, a mechanism that underlies its therapeutic benefit in a range of autoimmune and inflammatory disorders. Additionally, its antifibrotic properties help reduce the dermal deposition of glycosaminoglycans (mucins), thereby alleviating skin induration and improving mobility. As used in systemic sclerosis, IVIG is generally well tolerated with minimal adverse effects, making it a promising option for long-term management [65]. However, its reported effectiveness is based on a small number of cases; therefore, its true benefits remain uncertain. Furthermore, given the high cost of IVIG, careful evaluation of its cost-effectiveness, along with the identification of patient profiles most likely to respond, is warranted before its routine use in this rare condition can be recommended. Implementing guidelines to optimize the use of IVIG can help manage costs while ensuring appropriate patient selection [66].

MTX is a folate antagonist that inhibits dihydrofolate reductase, thereby blocking purine and pyrimidine synthesis, which are essential for DNA replication and cellular proliferation [67]. In addition to its classical antifolate activity, MTX exerts multiple non-dihydrofolate reductase-mediated effects, including modulation of oxidative stress, induction of cellular differentiation, epigenetic regulation via DNA and protein demethylation, histone acetylation, and suppression of proinflammatory cytokine production through adenosine-mediated pathways. Collectively, these mechanisms attenuate fibroblast proliferation, alter collagen metabolism, and downregulate inflammatory cascades, all of which have been implicated in SD pathophysiology. MTX modulates matrix metalloproteinase-1 and type I collagen expression in dermal fibroblasts, indicating a direct role in remodeling the dermal extracellular matrix [68]. The therapeutic effectiveness of MTX (MTX) in SD remains uncertain, with the majority of reports indicating minimal or nondurable clinical benefits.

Several cases have documented an absence of clinical improvement despite adequate treatment duration. Kyriakou et al. [22] reported no response after 5 months of MTX 15 mg/week in a female patient with long-standing type 1 diabetes, while Gracie and Whitaker [17] observed no benefit from low-dose MTX in a female with a 6-year disease history. Similarly, Mamadpur and Singh [16] followed five adults treated with MTX (alone or with PUVA) for up to 24 months, with no improvement or progression in all cases. In contrast, Doǧramaci et al. [39] reported moderate softening, reduced skin thickness, and improved range of motion in five adults after 3 months of MTX 15 mg/week, supported by histological improvement and no adverse effects. A retrospective multicenter study by Rongioletti et al. [3] described complete responses in two patients receiving MTX-based combination regimens, whereas Shazzad et al. [35] reported clinical improvement with MTX and PUVA in a single patient. However, Breuckmann et al. [53] found no improvement in seven patients after 6 months of MTX 25 mg/week despite concurrent physiotherapy. In another case, MTX was ineffective before tamoxifen initiation, which later led to a marked improvement [42].

Overall, these reports suggest that MTX monotherapy often yields suboptimal outcomes in patients with SD, whereas combination regimens—particularly phototherapy—may offer greater benefits. Variability in treatment responses may reflect differences in disease chronicity, severity, comorbidities, and concomitant interventions.

### 4.1. Limitations of the Study

First, most of the included publications were case reports or small case series with no well-designed randomized controlled trials available. Consequently, the true effectiveness of treatments could not be robustly evaluated, and the generalizability of the findings is limited. Moreover, reliance on such study designs increases the likelihood of reporting bias, as cases with positive or unusual outcomes are more likely to be published, potentially overestimating treatment efficacy. Hence, the treatment success rates presented in this review should be viewed with caution, as those rates may be attributed to reporting bias rather than true efficacy rates across a broader patient population. Second, there was considerable heterogeneity across studies in patient characteristics, treatment regimens, and follow-up durations, which precluded a direct comparison of clinical responses. Third, although combination therapy appeared to achieve better clinical outcomes than monotherapy in some reports, these observations were not derived from appropriately designed comparative studies, and adverse events were not consistently evaluated or reported in relation to monotherapy. Future multicenter prospective cohort studies utilizing standardized reporting of adverse events are warranted to fully understand the safety and tolerability of each intervention. Fourth, the literature search was restricted to articles published in English, which may have led to language bias and omission of relevant studies published in other languages. Finally, there are currently no standardized or consensus guidelines for the diagnosis and management of SD. This underscores the need for collaborative efforts between dermatology, rheumatology, endocrinology, and rehabilitation specialists to develop multidisciplinary management strategies and evidence-based treatment regimens.

### 4.2. Emerging Treatment Options

Current evidence suggests that the most reproducible clinical responses to refractory SD are achieved with phototherapy—particularly PUVA and UVA-1—and electron beam radiotherapy, as supported by multiple case reports and series [6,8,15,17,22,35,46,47,52]. Among the emerging therapeutic approaches, Janus kinase (JAK) inhibition and tranilast currently offer the strongest translational rationale for SD. JAK inhibitors’ biological plausibility is supported by evidence of JAK–STAT pathway crosstalk with profibrotic signaling pathways, as well as clinical experience demonstrating efficacy in other fibrosing dermatoses [69]. Preclinical studies have shown that JAK inhibitors exert antifibrotic effects by blocking transforming growth factor-β–mediated pathways. In bleomycin-induced mouse models, JAK1 and JAK2 inhibition effectively reduced skin fibrosis and improved vascular manifestations [70,71]. Lin et al. [72] reported the case of a 39-year-old female diagnosed with scleredema (unspecified type). The disease was refractory to multiple treatments including topical and intralesional corticosteroids, systemic corticosteroids, and cyclosporine. The patient was subsequently treated with oral tofacitinib, a non-selective JAK inhibitor, for 6 weeks. The patient experienced a rapid reduction in neck skin thickening and resolution of tenderness, with MRI confirming regression of the lesions. This finding supports the notion that JAK–STAT signaling contributes to fibroblast activation and collagen production in scleredema. Similarly, in localized scleroderma and systemic sclerosis, JAK inhibitors have shown beneficial clinical responses, including improvements in skin fibrosis, inflammation, and functional outcomes, with additional evidence of antifibrotic effects through modulation of the PI3K/Akt/mTOR signaling pathway [73,74]. Collectively, these findings support the cautious exploration of JAK inhibitors in SD, preferably in prospective controlled studies that incorporate rigorous safety monitoring. Another possible favorable agent is tranilast, an antifibrotic and anti-mast cell agent with anti-transforming growth factor-β activity. Tranilast has been reported to induce clinical improvement in three SD cases [29], and a dedicated clinical trial on SD has been registered. Thus, tranilast is one of the few agents with both preliminary clinical evidence and a formal trial pathway for this rare condition. Until more rigorous evidence is available, the use of these emerging therapies should be confined to structured protocols or clinical trials with shared decision-making that balances potential benefits against known and unknown risks.

Future investigations should aim to create standardized treatment protocols by performing prospective multicenter cohort studies, which would allow therapeutic modalities to be compared using similar response measures. To support this, we need randomized controlled trials to evaluate the effectiveness and safety of new therapies, such as JAK inhibitors and tranilast. Collaborative registries could also help collect data on this rare disorder. Additionally, to develop expert consensus guidelines on the diagnosis, assessments of responses, and long-term management of scleredema diabeticorum, a Delphi process potentially involving dermatologists, endocrinologists, rheumatologists, and experts in rehabilitation could provide a much-needed option for harmonization in this area.

## 5. Conclusions

This scoping review emphasized significant variability in the management of SD, with phototherapy (notably PUVA and UVA-1) and intravenous immunoglobulin displaying possible benefits, albeit with limited evidence from primarily small case reports and series. MTX has demonstrated limited and inconsistent efficacy, with better responses observed when combined with phototherapy. Evidence for other therapies, including colchicine, electron beam radiation, tranilast, and JAK inhibitors, remains preliminary but suggests potential benefits in selected refractory cases. The overall evidence base was constrained by the reliance on case reports, small series, variable treatment regimens, and inconsistent outcome reporting. There is an urgent need for well-designed prospective studies to define optimal therapeutic strategies, identify predictors of treatment response, and establish standardized management guidelines. Collaborative multidisciplinary approaches are essential to improve patient outcomes in this rare and challenging condition.

## Figures and Tables

**Figure 1 diseases-13-00346-f001:**
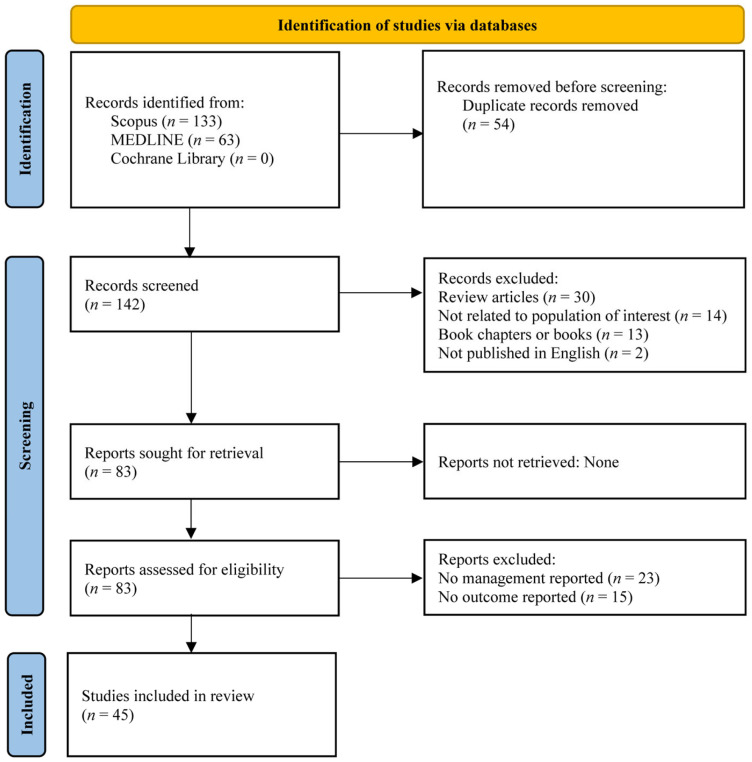
PRISMA 2020 flow diagram for study selection. MEDLINE, Medical Literature Analysis and Retrieval System Online; PRISMA, Preferred Reporting Items for Systematic Reviews and Meta-Analyses.

**Table 1 diseases-13-00346-t001:** Summary of included studies and reported clinical outcomes.

Ref. (Year)	Country	Patient Characteristics	Disease Onset/Duration	Treatment Details	Clinical Response
[15] (2025)	Spain	3 patients (2 females, 1 male); age 68–81; 2 with long-standing type 2 DM + MGUS, 1 with type 1 DM	Duration: 2–5 years	Low-dose UVA-1 (25 sessions, 3×/week, starting at 5 J/cm^2^, up to 20 J/cm^2^)	1 complete resolution; 2 with adequate improvement; no recurrence (3–24 months)
[16] (2025)	India	Five adults (4 males, 1 female), ages 51–62, all with long-standing uncontrolled type 2 diabetes (duration 10–25 years); 2 with diabetic retinopathy	Skin thickening duration: 2–15 months	Methotrexate (*n* = 3), methotrexate + PUVA (*n* = 1), topical tacrolimus (*n* = 1), colchicine (*n* = 1); all with diabetes control	No improvement in any case after 15–24 months of follow-up; progression of skin thickening in all
[17] (2024)	USA	A 57-year-old female with T1DM; 6-year history of SD; complications include adhesive capsulitis and multiple shoulder surgeries	Chronic (6 years); initial diagnosis via biopsy; recurrent progression despite treatments	Initial PUVA (3 months, ineffective); multiple courses of electron beam radiation (2016–2020; 24 Gy in 12 fractions and 12 Gy in 6 fractions); later methotrexate (low dose, stopped due to side effects); physical therapy, dry needling, and percussive devices were also tried.	Temporary improvement in skin texture and mobility after radiation; short-lived complete response with the latest course; no benefit from methotrexate or physical interventions; progression led to ineligibility for further electron therapy due to anterior neck involvement.
[18] (2024)	UK	A 68-year-old male with type 2 DM and diabetic retinopathy; HbA1c 10–11% over many years	3-year history of progressive skin thickening	Referred for the endocrine management of diabetes; phototherapy and physical therapy are recommended.	After 6 months: neck pain resolved, but the degree of skin thickening remained unchanged; HbA1c improved to 9.1%
[19] (2024)	Sri Lanka	A 49-year-old male with type 2 DM since 2013; poorly controlled; BMI 36.3 kg/m^2^; multiple diabetic complications (retinopathy, nephropathy, neuropathy, ED)	The patient has experienced a gradual onset of skin symptoms starting in 2018.	Improved glycemic control with oral antidiabetics (metformin, gliclazide, empagliflozin, sitagliptin); supportive care	HbA1c improved to 6.8% after 4 months; however, no clinical improvement of SD was observed at the 6-month follow-up.
[5] (2024)	Pakistan	A 48-year-old female with type 2 DM for 15 years; poorly controlled (HbA1c 9.3%); microvascular complications (neuropathy, nephropathy); history of PCI and depression	Gradual onset; duration not precisely stated	Referred for NB-UVB phototherapy; physiotherapy for restricted shoulder and neck movement; diabetes counseling and multidisciplinary care	After 3 months, improvement in symptoms and functionality was noted.
[20] (2022)	India	A 13-year-old female with poorly controlled type 1 diabetes mellitus, a history of DKA and diabetic retinopathy.	The initial episode lasted 6 months; recurrence at 8 months after resolution.	Split-mix insulin regimen (Mixtard), glycemic control, symptomatic treatment	The first episode resolved after 2 months of glycemic control; recurrence improved after restarting Mixtard; residual facial induration noted, continues follow-up.
[21] (2021)	South Korea	A 53-year-old male with type 1 DM (poorly controlled), hypertension, obesity, and newly diagnosed monoclonal gammopathy (IgG κ); no recent infections	1-year history of progressive skin hardening	Initially treated with NB-UVB phototherapy and methotrexate (10–20 mg/week) for 4 months (worsened). The patient was then treated with monthly IVIG (2 g/kg over 5 days) combined with NB-UVB.	After 4 cycles of IVIG: significant improvement in skin and mobility; after 10 cycles: continued improvement without side effects; ongoing monthly IVIG with NB-UVB yielded sustained response
[22] (2021)	Greece	A 55-year-old female; type 1 DM for 35 years; HbA1c 6.5% (controlled initially); newly diagnosed hypertension	Long-standing diabetes; symptom onset not explicitly stated	Treated sequentially with cyclosporine 150 mg twice daily for 3 months, followed by methotrexate 15 mg weekly for 5 months. Prednisolone 20 mg daily was given for 1 month but discontinued due to diabetes. Electron-beam radiation was subsequently administered. Lastly, colchicine and doxycycline were prescribed.	No response to cyclosporine or methotrexate. Prednisolone resulted in partial improvement but was stopped. Electron-beam radiation led to partial improvement. Minimal improvement observed with colchicine and doxycycline.
[23] (2021)	Spain	A 62-year-old male with well-controlled type 2 DM, HT, and dyslipidemia (Ten additional patients were treated with UVA-1 with good outcomes, but diabetes status was not reported).	Several months of dysphagia	Prednisone (tapering dose starting at 40 mg/day), methotrexate (15 mg/week for 4 months), methylprednisolone (6 pulses of 500 mg), intravenous immunoglobulin (3 g/week for 3 cycles), cyclophosphamide, followed by UVA-1 phototherapy (28 sessions; cumulative dose 291.09 J/cm^2^).	No response to systemic therapies; UVA-1 led to improved dysphagia, mobility, and neck skin stiffness
[24] (2020)	Spain	A 43-year-old female with type 2 DM, anterior uveitis, and past pancreatitis, presented with indurated, pruritic plaques and submaxillary gland inflammation	Duration not specified.	Initially treated with topical corticosteroids and antihistamines (ineffective); then systemic corticosteroids (prednisone 0.5–1 mg/kg/day, tapered to 2.5 mg/day)	Clinical improvement was maintained without relapse for 2 years; the diagnosis was revised from scleredema to IgG4-related disease based on lymph node biopsy and elevated serum IgG4
[25] (2020)	Spain	11 diabetic patients (3 with type 1 DM, 8 with type 2 DM); 7 were females; mean age 59 years (range: 43–89); 91% obese; 91% had DM ≥ 10 years (median: 15 years); mean HbA1c: 9.3% (range: 7.7–14.4); 91% had chronic DM complications; 91% treated with insulin; 40% had high insulin requirements (≥1 IU/kg)	Chronic course associated with long-standing diabetes mellitus; SD location: generalized (36%), multiple areas (36%), back only (27%)	5 patients (45%) treated with PUVA and topical psoralens; 2 patients improved with weight reduction and metabolic control; all insulin-dependent patients advised to avoid injecting insulin into affected areas	PUVA: all 5 had partial improvement; recurrence in 2; 2 patients improved with non-pharmacologic measures; 1 elderly patient avoided DM decompensation by altering insulin injection sites
[26] (2019)	Japan	A 47-year-old female with type 2 diabetes (HbA1c 7.6%)	3-month history of posterior cervical induration	18 sessions of topical hyaluronidase injection over 12 weeks	Marked improvement; sustained at 24-month follow-up
[27] (2018)	USA	A 54-year-old female with poorly controlled type 2 DM (HbA1c 12.7%) and IgG lambda monoclonal gammopathy; BMI 35.9 kg/m^2^	Progressive over at least 1 year	Diabetes management with insulin and oral hypoglycemics; dysphagia addressed through compensatory swallowing strategies and dietary modification; supportive ophthalmologic care	No clinical improvement in dysphagia or skin changes after 1 year; weight and glycemic control remained suboptimal (HbA1c 12.7%)
[28] (2018)	Austria	A 56-year-old Caucasian male; newly diagnosed type 2 DM (HbA1c 10%, BMI 37); no prior CTD history; progressive SAB	6 months before presentation	Initially treated with methylprednisolone (1.5 mg/kg body weight), tapered slowly. Due to disease progression and low compliance, therapy was switched to high-dose intravenous immunoglobulin (2 g/kg), which halted further progression. Insulin therapy was initiated. Subsequently, a 2-month rehabilitation program included ultrasound therapy (1 MHz, 1.5 W/cm^2^, 10 min per session, 17 sessions), manual lymphatic drainage (30 min per session, 9 sessions), and physiotherapy (30 min per session, 3 sessions), with home-based exercises prescribed daily.	Disease progression halted with IVIG. Functional improvement was observed after rehabilitation, including increased cervical spine mobility, decreased pain, and improved scores in 5 out of 8 SF-36 domains. BMI remained unchanged.
[29] (2018)	China	Three adults (2 males aged 45 and 49, 1 female aged 62) with type 2 diabetes and chronic progressive skin thickening	Chronic (up to 5 years)	Tranilast 0.3 g/day for 3 months	All showed skin softening and reduced dermal thickness (0.6–2.0 mm); improved mobility and daily function.
[30] (2017)	Italy	A 55-year-old female with type 2 DM (10-year duration); HbA1c 10.1%, BG 350 mg/dL; hypertriglyceridemia (1333 mg/dL)	Gradual onset over 13 months	Treated with oral corticosteroids (prednisone 0.5 mg/kg/day) and insulin for 6 weeks	Satisfactory improvement in skin lesions; surgical excision is considered if relapse occurs
[31] (2017)	USA	A 34-year-old female with type 1 DM (18 yrs), poor control; 60 M with diabetes	2 years; NA (response in 1 month)	IVIG: 34F received 2 g/kg every 4 weeks ×11 cycles; 60 M received 1.35 g/kg over 3 days (3 cycles q 6 weeks, then 5 cycles q 4 months)	34F: Improved ROM/induration after 2 cycles; relapsed off treatment; 60 M: Well tolerated with response observed within 1 month
[32] (2016)	USA	A 61-year-old male with type 2 DM (HbA1c 7.5%), hypertension, hypothyroidism, obesity (BMI 40.6); 4-year history of skin thickening, neck pain, and occipital headaches	Gradual progression over 4 years	Lifestyle modification; weight loss; improved glycemic control; no specific medical therapy initiated	At 4-year follow-up: ~16 kg weight loss, controlled diabetes, reduced neck pain/stiffness, and improved skin induration
[33] (2016)	Tunisia	A 36-year-old male, newly diagnosed with diabetes mellitus	Acute onset (10 days)	Prednisone 1 mg/kg/day and insulin	Partial improvement in edema and skin induration at 3 weeks; no progression after prednisone tapering
[6] (2016)	Turkey	A 54-year-old female with type 2 DM for 17 years; on insulin for 6 years; comorbidities include hypertension, coronary artery disease, diabetic neuropathy, and non-proliferative retinopathy; BMI 35 kg/m^2^	Gradually developing plaques; exact duration not specified	Intensive insulin therapy, gabapentin for neuropathy, and local PUVA therapy	After 2 months of PUVA: partial resolution of erythema and edema, improved mobility, and softer skin on the upper back
[34] (2015)	Portugal	A 30-year-old male with 2-year history of progressive skin induration; newly diagnosed primary Sjögren’s syndrome (ANA 1:1280, anti-SSA+, Schirmer’s test < 5 mm)	2-year history at initial presentation; symptoms progressed before diagnosis	Treated with hydroxychloroquine 400 mg daily	After 1 month: improvement in joint symptoms and stabilization of skin changes
[3] (2015)	Multicenter (France, Italy, Germany, Macedonia, Belgium, Switzerland)	44 patients (26 males, 18 females), mean age 53.8 years; 30 patients with diabetes mellitus	Chronic course; mean follow-up duration 32.2 months (range 1–185 months)	Phototherapy (either UVA-1 or PUVA) was the therapeutic modality most frequently associated with positive responses, which were usually partial. Systemic corticosteroids produced a complete response in one patient, and other immunosuppressant agents were rarely used (colchicine and other immunosuppressants were selective for myeloma). Colchicine was ineffective (*n* = 2), and simply optimizing diabetic therapy (*n* = 1) was associated with complete remission; another patient improved after optimizing their diabetes management. Selected combined regimens led to complete responses (e.g., extracorporeal photopheresis + physiotherapy; corticosteroids + imatinib in CML), while other, more complicated multi-agent treatment regimens did not lead to a complete response due to UVA-1–related problems. Responders were heterogeneous; phototherapy should be attempted first, and optimal glycemic control and physiotherapy are important components of care.	
[35] (2015)	Bangladesh	A 54-year-old male with a 16-year history of diabetes mellitus (insulin-treated; HbA1c 8.1%), hypertension for 6 years, and diabetic retinopathy.	1.5 years prior to admission; progressive	Intensified insulin regimen, followed by methotrexate 5 mg weekly for 2 months (partial clinical improvement), then escalated to methotrexate 10 mg weekly combined with PUVA therapy twice weekly for 1 year.	Improved shoulder abduction; decreased neck thickening; improved arm range of motion, daily activities, and ease of breathing
[36] (2014)	Italy	A 55-year-old male with type 1 diabetes, poor glycemic control, proliferative retinopathy, and ischemic heart disease	Progressive skin hardening; duration not clearly stated	Frequency Rhythmic Electrical Modulation System (FREMS) therapy (5 series over 15 months)	Significant clinical improvement (Barthel Index from 10 to 17), improved mobility and breathing, though no changes in imaging or histopathology; sustained benefits for over 1 year after final treatment
[37] (2014)	Taiwan	A 50-year-old female with type 2 DM for 20 years (on insulin for 10 years); 5-year history of progressive back induration and heat intolerance due to anhidrosis	5 years	Oral allopurinol 100 mg/day	After 8 months: slight improvement in erythema and induration; persistent anhidrosis due to eccrine gland loss
[38] (2014)	Saudi Arabia	A 45-year-old female with type 1 DM for 14 years; poorly controlled (HbA1c 10–11.5%); BMI 42.6 kg/m^2^; preproliferative diabetic retinopathy	6-month history of generalized indurated edema and progressive shoulder limitation	Improved insulin therapy to optimize glycemic control	After 1 year: HbA1c reduced to 8%, but no clinical improvement in skin lesions
[39] (2012)	Turkey	Five adults (3 females aged 52–74, 2 males aged 40 and 56) with type 2 diabetes mellitus (duration 2–25 years); all with progressive skin thickening; some with chronic renal failure or misdiagnosed as morphea	Chronic (2–10 years)	Methotrexate 15 mg/week SC + folic acid 1 mg/day (6 days/week)	After 3 months: moderate reduction in skin thickness, softening, and improved range of motion in all; histology showed decreased collagen and edema; no adverse effects
[40] (2012)	Thailand	A 53-year-old female, poorly controlled DM	2 years	PUVA (2×/week) + colchicine 1.8 mg/day (added after 10th session)	Significant improvement after 20 sessions; 50% softening at 20 sessions, 80% at 40 sessions; dermal thickness reduced; no longer needed analgesics at night
[41] (2011)	Taiwan	A 51-year-old male; medical history: diabetes mellitus (20 years), hypertension (3 years), chronic renal failure on hemodialysis (5 years)	Scleredema plaque: 20 years duration; diffuse pruritic eruption (4 months)	Allopurinol 100 mg/day (after negative HLA-B*58:01 test), primarily for acquired reactive perforating collagenosis	After 14 months, almost all papules and nodules resolved; a coincidental improvement in SD was observed, suggesting the potential benefit of allopurinol’s antioxidant properties
[8] (2011)	Spain	A 53-year-old white male with type 2 diabetes for 20 years; HbA1c < 7%; comorbid incipient nephropathy, retinopathy, obesity (BMI 30.3)	Skin changes present for ~10 years prior to diagnosis	Physiotherapy and PUVA therapy (UVA dose 120 J/cm^2^); UVA-1 was recommended but unavailable	Improvement in mobility of the back and shoulders; disappearance of erythema; skin softening after 2 months
[42] (2010)	Canada	Case 1: A 61-year-old female, type 2 DM with 2-year progressive trunk/shoulder skin thickening; Case 2: A 54-year-old female, type 2 DM with 5-year back/palm involvement	Case 1: 2-year onset; Case 2: 5 years	Tamoxifen 20 mg BID, reduced to QD; Case 1 failed MTX and D-penicillamine prior.	Case 1: Marked skin softening after 4 years, relapse on dose reduction; Case 2: Skin/palmar softening and ROM improvement after 4–18 months, relapse off treatment
[43] (2010)	Qatar	A 48-year-old male, type 2 DM for 20 years, irregular treatment; woody induration of neck, upper back, and arms	3 years	Insulin, diet, and exercise for glycemic control	Mild softening of lesions after 2 months
[44] (2010)	Thailand	Case 1: a 65-year-old male, T2DM ×10 years, poor glycemic control; Case 2: a 40-year-old male, T2DM ×2 years	Case 1: 10 yrs; Case 2: 1 yr	Both received medium-dose UVA-1 (60 J/cm^2^), 30–40 sessions	Case 1: >75% softening, sustained 2 yrs; Case 2: Improved ROM after 8 sessions, relapse at 10 months, responded to second course
[45] (2009)	UK	A 41-year-old male with poorly controlled type 1 diabetes for 29 years and microvascular complications	Diffuse skin tightness and shoulder limitation; exact duration not specified.	Conservative treatment: emollients and topical steroids for 6 months	Minimal improvement
[46] (2009)	South Korea	A 43-year-old male with newly diagnosed DM and rectal carcinoid tumor; scleredema symptoms ×10 years (neck), ×3 years (back)	10 years (neck), 3 years (back)	Electron beam radiation: 20 Gy total (2 Gy/day)	Improved pain, erythema, and range of motion; no change on computed tomography imaging; sustained symptom relief
[47] (2008)	Netherlands	3 patients (2 males, 1 female; ages 51–66) with long-standing, poorly controlled diabetes mellitus	Not specified; severe induration with limited neck/shoulder mobility	Medium-dose UVA-1 (35–60 J/cm^2^/session; cumulative dose ~1400–1460 J/cm^2^)	All improved: 1 excellent, 2 good responses; skin softened, mobility improved, durometry confirmed progress
[48] (2008)	Qatar	A 55-year-old male, type 2 DM for 10 years, skin thickening on thighs, neuropathy, and respiratory symptoms	Skin lesions 2 years after DM diagnosis; progressive over several years	Not specified	Worsened with respiratory failure and died suddenly at home
[49] (2007)	Germany	3 patients (A 55-year-old male, a 57-year-old female, and a 58-year-old female) with type 2 DM and progressive skin thickening	2 years (55 M), 20 years (57F), 2 months (58F)	55 M: UVA-1; 57F: UVA-1 + IV clindamycin; 58F: IV penicillin	55 M: Marked improvement in 4 weeks; 57F: Slight improvement; 58F: Marked improvement after 2 months
[50] (2006)	France	4 patients (1 male, 3 females) with long-standing type 1 DM and Buschke’s scleredema	Not specified (implanted 1994–2004)	Intraperitoneal insulin via an implantable pump	Dramatic improvement of skin induration and glycemic control (HbA1c decreased from 9.3% to 7.9%); stable/decreased microvascular complications
[51] (2006)	France	A 44-year-old male, type 1 DM for 17 years, poor control (HbA1c 8–9.6%), retinopathy, restrictive lung function	Progressive over years	PUVA (15 sessions, 73.5 J) + physiotherapy	No skin improvement; physiotherapy continued
[52] (2006)	Japan	A 53-year-old male (DM for 8 years) and A 52-year-old female (DM for 13 years) with scleredema, duration of 8 years	Not specified	Oral PUVA therapy (total UVA dose: 128 J/cm^2^ and 101 J/cm^2^, respectively); cyclosporine trial in 53 M ineffective	Both showed marked clinical and histological improvement; cyclosporine had no benefit.
[53] (2005)	Germany	7 patients (5 males, 2 females), mean age 56, poorly controlled insulin-dependent DM, obese (mean BMI 37.3)	Mean 26 months (range 6–48)	Oral methotrexate 25 mg/week + physiotherapy for 6 months	No clinical improvement or ultrasound/densitometric change; some reported improved mobility only
[54] (2005)	USA	A 40-year-old male, type 1 DM since childhood, with neuropathy and retinopathy; woody induration on neck, back, and hands; unable to form a fist.	Slowly progressive	Aminobenzoate 500 mg TID, colchicine 0.6 mg BID, Dimethyl sulfoxide (DMSO) gel, glucose control	Improved mobility of hands and back; skin softening noted
[55] (2005)	Japan	A 58-year-old male with a 15-year history of type 2 DM, retinopathy, and long-term use of an electric massaging chair	A several-year history of indurated plaques on the posterior neck and upper back	No treatment given due to absence of symptoms	Stable without progression; mechanical stress suspected as a contributing factor

Note. BMI, body mass index; CTD, connective tissue disease; DKA, diabetic ketoacidosis; DM, diabetes mellitus; FREMS, Frequency Rhythmic Electrical Modulation System; HbA1c, glycated hemoglobin; HT, hypertension; IVIG, intravenous immunoglobulin; MGUS, monoclonal gammopathy of undetermined significance; MTX, methotrexate; NB-UVB, narrow-band ultraviolet B; PCI, percutaneous coronary intervention; PUVA, psoralen plus ultraviolet A; Ref, reference; ROM, range of motion; SC, subcutaneous; SD, scleredema diabeticorum; T1DM, type 1 diabetes mellitus; T2DM, type 2 diabetes mellitus; UVA, ultraviolet A; UVB, ultraviolet B.

## Data Availability

All data supporting the results are included in this article and its Appendix A.

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
