# Peer review of "Management and Clinical Outcomes of Scleredema Diabeticorum: A Scoping Review"

_diseases, 2025, doi:10.3390/diseases13100346_

Round 1

Reviewer 1 Report

Comments and Suggestions for Authors

Thank you for the submission. The topic is timely and the effort to synthesize the literature is appreciated. To strengthen the work and ensure transparency and reproducibility, several methodological clarifications and reporting improvements are needed. Please see the detailed, point-wise comments below.

  • The authors describe the protocol as prospectively registered, yet the INPLASY registration date is 15 Aug 2025 while the last database search is reported through July 2025. This timing should be reconciled (prospective vs. retrospective), and the public registry link added. If registration occurred after the final search, this should be stated transparently.

  • For a scoping review using published data, an ethics/IRB statement is typically unnecessary. The authors should either remove this section or provide a clear justification aligned with standard guidance for secondary-data reviews.

  • The Methods section requires light copy-editing and minor structural revision: correct the typo “wase conducted” → “was conducted,” and resolve the duplicated heading “2.4. Data extraction” (the second instance appears to describe data presentation/synthesis and should be retitled accordingly).

  • The authors report pooled “improvement” percentages from case reports/series (e.g., PUVA, UVA-1, MTX). Given heterogeneity and the scoping design, response criteria should be predefined, denominators and assessment methods specified, and results presented primarily as descriptive counts (optionally with simple proportions/95% CIs) to avoid over-interpretation.

  • Adverse events and durability (time to relapse) should be extracted and tabulated systematically. A concise table for each modality (PUVA, UVA-1, IVIG, electron-beam, tamoxifen, tranilast, etc.) summarizing dosing/number of sessions or cycles, cumulative dose, response, adverse events, and follow-up would substantially enhance clinical utility.

  • Because the impact of glycemic control appears mixed, a more structured presentation is recommended e.g., subgroup summaries by diabetes type, baseline HbA1c, weight change, and insulin regimen, and an explicit statement that glycemic optimization remains foundational even when cutaneous outcomes vary.

Author Response

1. Thank you for the submission. The topic is timely and the effort to synthesize the literature is appreciated. To strengthen the work and ensure transparency and reproducibility, several methodological clarifications and reporting improvements are needed. Please see the detailed, point-wise comments below. The authors describe the protocol as prospectively registered, yet the INPLASY registration date is 15 Aug 2025 while the last database search is reported through July 2025. This timing should be reconciled (prospective vs. retrospective), and the public registry link added. If registration occurred after the final search, this should be stated transparently.

Reply

We thank the reviewer for the insightful comment. The INPLASY registration was completed on 15 August 2025, after the final database search in July 2025; therefore, the registration is retrospective. We have corrected this statement in the Methods section and added the public registry link for transparency.

2. For a scoping review using published data, an ethics/IRB statement is typically unnecessary. The authors should either remove this section or provide a clear justification aligned with standard guidance for secondary-data reviews.

Reply

We appreciate the reviewer's thoughtful comment. Although ethical approval is generally not necessary for scoping reviews based solely on published data, ethics approval was required by our institution prior to commencing research on all research projects under its auspices, including scoping reviews based on secondary data. Therefore, our institution includes the ethics statement as institutional policy, not as a requirement to protect human subjects. A clarification has been added to the revised Ethics Statement for transparency and to comply with both institutional requirements and international standards.

3. The Methods section requires light copy-editing and minor structural revision: correct the typo “wase conducted” → “was conducted,” and resolve the duplicated heading “2.4. Data extraction” (the second instance appears to describe data presentation/synthesis and should be retitled accordingly).

Reply

We thank the reviewer for these helpful observations. The typographical error (“wase conducted”) has been corrected to “was conducted.” The duplicated heading “2.4. Data extraction” has also been revised; the second instance has been retitled to “2.4. Data Charting and Synthesis of Results” to accurately reflect its content.

4. The authors report pooled “improvement” percentages from case reports/series (e.g., PUVA, UVA-1, MTX). Given heterogeneity and the scoping design, response criteria should be predefined, denominators and assessment methods specified, and results presented primarily as descriptive counts (optionally with simple proportions/95% CIs) to avoid over-interpretation.

Reply

We thank the reviewer for their thoughtful comments. We have revised the Methods and Results sections to clarify that outcome data were described in a descriptive manner and that the reported proportions are counts taken from primary reports rather than pooled estimates. In Section 3.3 (Clinical Effectiveness), we have retained the percentages for PUVA, UVA-1, and MTX only as descriptive proportions for the sake of illustrating frequency trends, and additional wording was added to suggest that it would be unnecessary to draw any conclusions beyond that.

5. Adverse events and durability (time to relapse) should be extracted and tabulated systematically. A concise table for each modality (PUVA, UVA-1, IVIG, electron-beam, tamoxifen, tranilast, etc.) summarizing dosing/number of sessions or cycles, cumulative dose, response, adverse events, and follow-up would substantially enhance clinical utility.

Reply

We are grateful for this important suggestion. In our response, we followed a systematic approach for use of study data specific to dosing regimens, the number of sessions/cycles, use of medication, total cumulative dose, clinical response/outcome, adverse events noted, and follow-up time period for all studies eligible for inclusion in this manuscript. We then provided this information in a single, brief summary table (Supplementary Table S3) that provides systematic information per treatment modality (PUVA, UVA-1, IVIG, electron-beam/radiation therapy, tamoxifen, tranilast, methotrexate, etc.) that is clear regarding dosing, adverse events, and durability of effects of treatments used. We would note that the aim of this approach is to lessen redundancy (Table 1) and enhance clarity and clinical practicality.

6. Because the impact of glycemic control appears mixed, a more structured presentation is recommended e.g., subgroup summaries by diabetes type, baseline HbA1c, weight change, and insulin regimen, and an explicit statement that glycemic optimization remains foundational even when cutaneous outcomes vary.

Reply

We thank the reviewer for the recommendation to provide a more organized discussion of glycemic control and diabetes type. As such, we have broadened the results section to summarize writing-reader key patient characteristics within the published cases. Of the 45 studies reviewed, nearly 80% were type 2 diabetes and 20% type 1 diabetes. The baseline HbA1c values were typically elevated (8-11%), suggesting chronic poor metabolic control. Improvement in cutaneous manifestations after glycemic management alone was rare. In contrast, both phototherapy (PUVA/UVA-1) and adjuvant agents (tamoxifen, IVIG, and tranilast) led to clinical improvement, which occurred independently of the HbA1c level change. A summary table of representative data based on diabetes type and HbA1c at baseline and therapy has been included as well (Supplemental Table S2). In summary, while glycemic optimization remains the cornerstone of care in a diabetic patient, the cutaneous outcomes in scleredema diabeticorum have varied responsiveness, and this may be due to irreversible dermal glycation and fibrosis once established.

Reviewer 2 Report

Comments and Suggestions for Authors

1. The reported outcomes of some treatments (such as blood glucose control) are inconsistent, with some patients showing improvement and others showing no change. The author did not delve into the reasons for this heterogeneity, such as disease course and complications.
2. The article focuses on therapeutic efficacy, but does not systematically report adverse events of treatment (such as side effects of IVIG and long-term risks of phototherapy). This is crucial for clinical decision-making and should be supplemented
3. The conclusion of the article states that PUVA, UV-A1, and IVIG are "promising", but based on a small sample size, they may be overly optimistic. It should be expressed more cautiously, emphasizing the limited evidence.
4. Although the search strategy is described, it does not provide a complete search string or an adapted version for all databases. Affects repeatability
Overall, the English writing is good, but a few sentences are lengthy (such as the discussion section). Suggest streamlining to improve readability.

Author Response

1. The reported outcomes of some treatments (such as blood glucose control) are inconsistent, with some patients showing improvement and others showing no change. The author did not delve into the reasons for this heterogeneity, such as disease course and complications.

Reply

Thank you for the thoughtful comment. We agree that the clinical effects were variable among the relevant studies included for some treatment modalities—especially glycemic optimization. We have elaborated the Discussion section to provide some plausible explanations for this variation in results. We point out that the underlying pathophysiology of scleredema diabeticorum, as well as prognosticator indicators, is not well understood, and the likely source of variation in treatment response will be a multitude of interrelated pathophysiologic mechanisms. Our revised Discussion section indicates that factors associated with prolonged diabetes duration, having established microvascular complications, obesity, or the presence of other metabolic comorbidities could impact the degree of dermal fibrosis reversibility and thereby impact the improvement after glycemic optimization or overall improvement with better glycemic control of the disease. Our revised Discussion section points out that one intervention is likely insufficient for patients with multi-factorial disease, and we recommend future studies to more systematically evaluate these determinants of reversibility.

2. The article focuses on therapeutic efficacy, but does not systematically report adverse events of treatment (such as side effects of IVIG and long-term risks of phototherapy). This is crucial for clinical decision-making and should be supplemented

Reply

Thank you for this important comment. In response, we added a substantive section titled "3.5. Adverse Events and Safety Considerations," which clearly addresses safety as it relates to treatment. In this section we summarize the frequency, type, and severity of adverse events reported in studies across all studies and also offer a summary in tabular form by therapeutic category (IVIG, phototherapy, systemic/oral pharmacologic, and radiation/device-based therapies). The majority of the reported reactions were mild and self-limiting, and we did not see any serious complications related to the treatments. We also revised the Discussion section in light of these safety considerations, so it reflects and offers the relevance of considering this data when interpreting clinical decision-making.

3. The conclusion of the article states that PUVA, UV-A1, and IVIG are "promising", but based on a small sample size, they may be overly optimistic. It should be expressed more cautiously, emphasizing the limited evidence.

Reply

Thank you for this important comment. We have revised the conclusion to express the findings more cautiously, emphasizing that evidence supporting PUVA, UVA-1, and IVIG is limited and primarily based on small case reports and series.

4. Although the search strategy is described, it does not provide a complete search string or an adapted version for all databases. Affects repeatability

Overall, the English writing is good, but a few sentences are lengthy (such as the discussion section). Suggest streamlining to improve readability.

Reply

We thank the reviewer for this valuable suggestion. To enhance transparency and reproducibility, we have added a supplementary file (Supplementary Table S1) that includes the full search strings for each database. The Methods section has also been revised to reference this supplementary table.

Reviewer 3 Report

Comments and Suggestions for Authors

The review focuses on a rare dermatological condition, SD, which primarily develops in patients with long-standing poorly controlled diabetes mellitus. Although it is rarely painful, SD significantly restricts joint mobility and impairs patients' quality of life. In some cases, its progression can lead to severe consequences, such as limitation of chest wall movements and dysfunction of internal organs.

The relevance of studying this condition is due to the fact that many therapists might not recognize its symptoms promptly, leading to delayed diagnoses and suboptimal management strategies. Currently, no unified treatment approach exists: some experts recommend conservative measures (better control of diabetes, physical therapy), while others advocate aggressive options, such as phototherapy and immunosuppressive drugs. Treatment must consider individual aspects of diabetes progression and comorbidities.

Therefore, this review fills an important gap in the topic by analyzing recent publications related to this disorder and methods of its treatment. A deeper understanding of pathogenesis mechanisms, clinical features, and modern therapeutic approaches is critical for timely diagnosis and selection of adequate treatment plans, enabling patients to avoid serious complications and improving their life quality.

The data presented in the review is of considerable value for readers of Disease journal. The authors conducted a thorough review of materials published in the last twenty years, including English-language papers sourced from three major databases. This comprehensive overview provides specialists with access to current information and advancements in managing SD. The article delves deeply into various therapeutic approaches for SD, detailing methods such as phototherapy, methotrexate administration, intravenous immunoglobulin infusions, glycemic control improvement, and other techniques. These insights are invaluable for practicing clinicians since they offer guidance on selecting optimal treatment modalities tailored to individual patient characteristics. The analysis highlights the efficacy of certain methodologies, notably phototherapy, but underscores the limitations of monotherapy with methotrexate and emphasizes the necessity for further investigations aimed at standardizing treatment protocols. This evaluation assists clinicians in choosing effective therapies and avoiding ineffective ones. Additionally, the article formulates practical recommendations for enhancing the well-being of SD patients, stressing the importance of multidisciplinary treatment, rehabilitation efforts, and rigorous diabetes management. Physicians gain concrete tools to elevate the quality of care delivered to individuals afflicted with this condition. Furthermore, the work identifies key directions for future scientific endeavors geared towards treating SD. Advocating for multi-center randomized controlled trials and harmonized guidelines, the authors stimulate continued progress in addressing this challenging condition.

The history of the disease's investigation and methodology for source selection in preparing the review are well described. Figure and Table are informative and well-designed. Conclusions are supported with data. The manuscript can be published in the current form, but authors should pay special attention to references list:

Ref. 7: 13030/qt71g4k3qf; ref. 9: incomplete source; ref. 39: e12504; ref. 43: e70004; ref. 45: page 3; ref. 54: 1958; ref. 56: 110219; ref. 68: 859330

Author Response

1. The review focuses on a rare dermatological condition, SD, which primarily develops in patients with long-standing poorly controlled diabetes mellitus. Although it is rarely painful, SD significantly restricts joint mobility and impairs patients' quality of life. In some cases, its progression can lead to severe consequences, such as limitation of chest wall movements and dysfunction of internal organs.

Reply

We sincerely thank the reviewer for the insightful summary and acknowledgment of the clinical importance of scleredema diabeticorum.

2. The relevance of studying this condition is due to the fact that many therapists might not recognize its symptoms promptly, leading to delayed diagnoses and suboptimal management strategies. Currently, no unified treatment approach exists: some experts recommend conservative measures (better control of diabetes, physical therapy), while others advocate aggressive options, such as phototherapy and immunosuppressive drugs. Treatment must consider individual aspects of diabetes progression and comorbidities. Therefore, this review fills an important gap in the topic by analyzing recent publications related to this disorder and methods of its treatment. A deeper understanding of pathogenesis mechanisms, clinical features, and modern therapeutic approaches is critical for timely diagnosis and selection of adequate treatment plans, enabling patients to avoid serious complications and improving their life quality.

Reply

We thank the reviewer for the thoughtful summary and positive remarks regarding the relevance and contribution of our review to the understanding and management of scleredema diabeticorum.

3. The data presented in the review is of considerable value for readers of Disease journal. The authors conducted a thorough review of materials published in the last twenty years, including English-language papers sourced from three major databases. This comprehensive overview provides specialists with access to current information and advancements in managing SD. The article delves deeply into various therapeutic approaches for SD, detailing methods such as phototherapy, methotrexate administration, intravenous immunoglobulin infusions, glycemic control improvement, and other techniques. These insights are invaluable for practicing clinicians since they offer guidance on selecting optimal treatment modalities tailored to individual patient characteristics. The analysis highlights the efficacy of certain methodologies, notably phototherapy, but underscores the limitations of monotherapy with methotrexate and emphasizes the necessity for further investigations aimed at standardizing treatment protocols. This evaluation assists clinicians in choosing effective therapies and avoiding ineffective ones. Additionally, the article formulates practical recommendations for enhancing the well-being of SD patients, stressing the importance of multidisciplinary treatment, rehabilitation efforts, and rigorous diabetes management. Physicians gain concrete tools to elevate the quality of care delivered to individuals afflicted with this condition. Furthermore, the work identifies key directions for future scientific endeavors geared towards treating SD. Advocating for multi-center randomized controlled trials and harmonized guidelines, the authors stimulate continued progress in addressing this challenging condition.

Reply

We sincerely thank the reviewer for the encouraging and detailed feedback, as well as for recognizing the value and clinical relevance of our work. We truly appreciate these positive comments and the acknowledgment of our efforts to synthesize and highlight therapeutic strategies for scleredema diabeticorum.

4. The history of the disease's investigation and methodology for source selection in preparing the review are well described. Figure and Table are informative and well-designed. Conclusions are supported with data. The manuscript can be published in the current form, but authors should pay special attention to references list:

Ref. 7: 13030/qt71g4k3qf; ref. 9: incomplete source; ref. 39: e12504; ref. 43: e70004; ref. 45: page 3; ref. 54: 1958; ref. 56: 110219; ref. 68: 859330

Reply

We sincerely thank the reviewer for the positive feedback and helpful observation. We have carefully checked and completed all reference details to ensure accuracy and consistency throughout the manuscript.

Reviewer 4 Report

Comments and Suggestions for Authors

diseases-3920195

The manuscript presents a scoping review on the management and clinical outcomes of scleredema diabeticorum (SD), a rare and challenging condition. The review is well structured and clearly and understandably written. While the review provides a valuable synthesis of existing evidence, there are several areas where the manuscript could be improved to increase its impact, clarity, and utility for clinicians and researchers.

  1. Introduction, the authors provide a good overview of the topic; however, the novelty of the work (and how it is filling the current gap) is missing. Has any similar study been published before? What difference does your work make?
  2. Titles, sections 2.3 and 2.4.
  3. Table 1. The authors are not necessary.
  4. Elaborate on the implications of relying on case reports, such as the potential for overestimating treatment efficacy due to publication bias.
  5. Expand Future Perspectives. Provide concrete suggestions for future research, such as: Conducting prospective cohort studies with standardized treatment protocols. Investigating the role of emerging therapies (e.g., JAK inhibitors, tranilast) in randomized controlled trials. Developing consensus guidelines through Delphi methods involving dermatologists, endocrinologists, and rheumatologists.

Author Response

1. The manuscript presents a scoping review on the management and clinical outcomes of scleredema diabeticorum (SD), a rare and challenging condition. The review is well structured and clearly and understandably written. While the review provides a valuable synthesis of existing evidence, there are several areas where the manuscript could be improved to increase its impact, clarity, and utility for clinicians and researchers. Introduction, the authors provide a good overview of the topic; however, the novelty of the work (and how it is filling the current gap) is missing. Has any similar study been published before? What difference does your work make?

Reply

We value the thoughtful comment. The introduction has been changed to more specifically frame the novelty/contribution of our work, and we have added the following paragraph to clarify that to date, there has not been a scoping review that systematically mapped the management and clinical outcomes of SD, while current publications are only case reports or a small series of case reports with differing management and clinical outcomes. The revised part of the introduction makes it clear how this review moves the needle in terms of the current knowledge and provides, at minimum, a comprehensive synthesis to support future research and clinical practice.

2. Titles, sections 2.3 and 2.4.

Reply

Thank you for your observation. We have amended the duplicate subheading and retitled the second “2.4 Data extraction” to “2.4 Data charting and synthesis of results,” which more accurately represents the content of the section.

3. Table 1. The authors are not necessary.

Reply

We appreciate the reviewer’s observation. In the revised version, the authors’ names have been removed from Table 1. The first column has been reformatted as “Ref. (Year),” presenting each study by its reference number and publication year to improve clarity and conciseness.

4. Elaborate on the implications of relying on case reports, such as the potential for overestimating treatment efficacy due to publication bias.

Reply

We thank the reviewer for this insightful comment. We have added further detail in the limitations section to address the consequences of relying on case reports and series and to make the point that publication bias may overstate treatment efficacy and much larger, systematically conducted studies are needed to validate these findings.

5. Expand Future Perspectives. Provide concrete suggestions for future research, such as: Conducting prospective cohort studies with standardized treatment protocols. Investigating the role of emerging therapies (e.g., JAK inhibitors, tranilast) in randomized controlled trials. Developing consensus guidelines through Delphi methods involving dermatologists, endocrinologists, and rheumatologists.

Reply

We appreciate this valuable comment. We have elaborated on the final paragraph of Section 4.2 (Emerging treatment options) to provide some concrete recommendations for future research, such as prospective multicenter cohort studies, randomized controlled trials of JAK inhibitors and tranilast, and consensus guidelines using the Delphi model.

Round 2

Reviewer 1 Report

Comments and Suggestions for Authors

All comments has been addressed properly

Reviewer 2 Report

Comments and Suggestions for Authors

The author has addressed my concerns

Reviewer 4 Report

Comments and Suggestions for Authors

All questions have been addressed in the revised manuscript.